# Genetic Variation in CCL18 Gene Influences CCL18 Expression and Correlates with Survival in Idiopathic Pulmonary Fibrosis—Part B

**DOI:** 10.3390/jcm9061993

**Published:** 2020-06-25

**Authors:** Canay Caliskan, Benjamin Seeliger, Benedikt Jäger, Jan Fuge, Tobias Welte, Oliver Terwolbeck, Julia Freise, Coline H. M. van Moorsel, Yingze Zhang, Antje Prasse

**Affiliations:** 1Department of Respiratory Medicine, Hannover Medical School and Biomedical Research in End-stage and Obstructive Lung Disease Hannover, German Lung Research Center (DZL), 30265 Hannover, Germany; canay.caliskan@stud.mh-hannover.de (C.C.); Seeliger.benjamin@mh-hannover.de (B.S.); fuge.jan@mh-hannover.de (J.F.); welte.tobias@mh-hannover.de (T.W.); freise.julia@mh-hannover.de (J.F.); 2Fraunhofer Institute for Toxicology and Experimental Medicine, 30625 Hannover, Germany; benedikt.jaeger@item.fraunhofer.de (B.J.); oliver.terwolbeck@item.fraunhofer.de (O.T.); 3Dept of Pulmonology, Interstitial Lung Diseases Center of Excellence, St Antonius Hospital, 3435 CM Nieuwegein, The Netherlands; c.van.moorsel@antoniusziekenhuis.nl; 4Department of Medicine and Human Genetics, University of Pittsburgh, Pittsburgh, PA 15261, USA; zhangy@upmc.edu

**Keywords:** idiopathic pulmonary fibrosis, antifibrotics, CCL18, interstitial lung disease

## Abstract

Idiopathic pulmonary fibrosis (IPF) is a progressive disease with high mortality. CC-chemokine ligand 18 (CCL18) is predictive of survival in IPF. We described correlation of CCL18 serum levels with the genotype of *rs2015086* C > T polymorphism the *CCL18*-gene, which was associated with survival in a pre-antifibrotic cohort (Part-A). Herein (Part-B), we aimed to validate these findings and to study the effects of antifibrotics. Two cohorts were prospectively recruited, cohort-A (*n* = 61, pre-antifibrotic) and cohort B (*n* = 101, received antifibrotics). Baseline CCL18 serum level measurement by enzyme-linked immunosorbent assay (ELISA, serially in cohort B) and genotyping of *rs2015086* was performed and correlated with clinical outcomes. The CT genotype was present in 15% and 31% of patients. These patients had higher CCL18 levels compared to the TT-genotype (cohort-A: 234 vs. 115.8 ng/mL, *p* < 0.001; cohort B: 159.5 vs. 120 ng/mL, *p* = 0.0001). During antifibrotic therapy, CCL18 increased (*p* = 0.0036) regardless of *rs2015086*-genotype and antifibrotic-agent. In cohort-A, baseline CCL18-cutoff (>120 ng/mL) and CT-genotype were associated with mortality (*p* = 0.041 and *p* = 0.0051). In cohort-B, the CCL18-cutoff (>140 ng/mL) was associated with mortality (*p* = 0.003) and progression (*p* = 0.004), but not the CT/CC-genotype. In conclusion, we validated the correlation between *rs2015086*-genotype and CCL18 serum levels, which was predictive of (progression-free)-survival in two prospective validation cohorts.

## 1. Introduction

Idiopathic pulmonary fibrosis (IPF) is the most common type of idiopathic interstitial pneumonias. It is characterized by a radiologic and histopathological “usual interstitial pneumonia (UIP)” pattern and dismal prognosis with a median survival of 3 years following diagnosis if untreated [1,2]. Prior to the Prednisone, Azathioprine, and N-Acetylcystein in IPF (PANTHER) trial [3], routine immunosuppressive treatment was used in IPF. With the introduction of antifibrotic therapy, the decline in lung function could be slowed with both nintedanib [4] and pirfenidone [5] and emerging data suggest prolongation of life-expectancy [6,7]. Despite antifibrotics and numerous agents being investigated at present, there is an unmet need for development of personalized treatment options considering the significant morbidity and mortality in IPF [8]. Many biomarkers have been associated with progression and prognosis in IPF and the foremost one is CC chemokine ligand 18 (CCL18), which is a marker of alternatively-activated macrophages (M2 macrophages) with profibrotic mechanisms of action [9,10]. The serum level of CCL18 has been shown to predict survival in IPF and systemic sclerosis, including in depth analyses from data of two randomized controlled trials [11,12,13,14,15,16]. There are two single nucleotide polymorphisms (SNPs) in the promotor region of *CCL18* gene, *rs2015086* and *rs712040* which are presumably functional in gene regulation [10,17].

In the first part (A) of this work [18], we described how *rs2015086* C > T polymorphism in the *CCL18* promotor region influences CCL18 serum levels. Heterozygous carriers of the C allele (CT-genotype) had higher concentrations of serum CCL18 levels, higher *CCL18* mRNA expression and decreased survival compared to homozygous carriers of the common T allele [18] in a Dutch IPF-cohort predating the introduction of antifibrotic therapy. In this second part (B) of the project, we recruited two German validation cohorts (one pre-antifibrotic from Freiburg; the other with antifibrotics from Hannover) to validate the correlation of CCL18 serum levels with the *rs2015086* genotype and describe the influence of antifibrotic therapy on CCL18 levels during disease.

## 2. Experimental Section

### 2.1. Patients and Clinical Data

Patients were prospectively included in this study if they had (i) a confident diagnosis according to the American Thoracic Society/European Respiratory Society guidelines [1], (ii) had available serum samples at baseline, and (iii) provided informed consent. For the validation cohort of this study, patient databases were screened at two German tertiary interstitial lung disease centers. In center A at Freiburg, we screened patients between 2001 and 2013, most of which received immunosuppressive treatment as suggested by guidelines [19] before the results of the PANTHER trial [3]; in center B at Hannover, we included patients between 2014–2018, most of whom received antifibrotic therapy. Baseline characteristics including age, gender, percentage of predicted forced vital capacity (%FVC) and diffusion capacity for carbon monoxide (%DLCO), body mass index (BMI) were recorded. Pulmonary function tests were performed according to the European Respiratory Society / American Thoracic Society standards using a bodyplethysmograph [20]. Progression-free survival was defined as combined endpoint of either decline in FVC of >10% from baseline or decline of DLCO > 15% from baseline or death. To allow for equal follow-up time, clinical outcomes (survival in cohort A; survival and progression-free survival in cohort B), were censored at 48 months in cohort A and 36 months in cohort B.

The study was conducted in accordance with the 1964 Declaration of Helsinki and its later amendments and biomaterial collection and use of retrospective data were approved by the local ethics committee and all patients signed informed consent prior to inclusion (Freiburg 47/06 March 10th 2006, Hannover, #2923-2015 and #2516-2014, 2 November 2015). The study has also been registered at the German Clinical Trials Register (DRKS00000017 on 5 September 2008) and previously at the local clinical trials registry of the University of Freiburg on 15 September 2006).

### 2.2. Serum Sampling

Blood samples were collected at initial diagnosis (cohort A) or prior to the antifibrotic therapy (cohort B) and from a subset of patients after 3, 6 and 12 months during the antifibrotic therapy. Blood samples were rested for 20 min before centrifugation. After centrifugation, all samples were frozen at −20 °C. Subsequently. the samples were stored at −80 °C until performance of the enzyme-linked immunosorbent assay (ELISA).

### 2.3. Serum CC18 Measurement by Enzyme-Linked Immunosorbent Assay (ELISA)

Serum-CCL18-concentrations were measured by ELISA using the DuoSet^®^ Development System ELISA Kit (R&D System, Minneapolis, MN, USA) according to the provided manufacturer’s protocol. All samples were measured at least in two different ELISAs. Each sample on one plate was measured in duplicate. The mean of all measurements was used in the final analysis.

### 2.4. Single Nucleotide Polymorphism (SNP) Genotyping

Details of initial description and analysis of SNPs of the CCL18 promotor region and choice of *rs2015086* is described in detail in *part A* of this study (REF PART A). One hundred twenty-six subjects were genotyped for the CCL18 promoter polymorphism *rs2015086* using the TaqMan SNP Genotyping assay-allelic discrimination method (Thermo Fisher Scientific, Waltham, MA, USA) and the StepOnePlus™ real-time PCR system (Thermo Fisher Scientific, Waltham, MA, USA).

### 2.5. Statistical Analysis

All data were expressed as median with interquartile ranges (IQR). A two-tailed *p*-value of <0.05 was considered statistically significant. Group differences were assessed using the Wilcoxon test. For analysis of (progression—free) survival, the median serum CCL18 level in the respective patient cohort was used as a cut-off and its influence was estimated by the Kaplan–Meier method with log-rank test for statistical significance. Cox proportional hazard model was used to calculate hazard ratios (HR) with adjustment for baseline variables including %FVC, age, sex, antifibrotic treatment and BMI. In an analogy to Oldham et al. *2015*, rs2015085-genotype-treatment interaction was tested with a multivariable cox regression model including the genotype (TT or CT/CC), antifibrotic agent and genotype-treatment interaction term [21,22]. Gene-treatment interaction was considered present if the *p*-value of the Wald z-statistic for the interaction term was <0.01 [22]. Statistical analysis and graph generation was performed using STATA V16 (STATA Corp LP, College Station, TX, USA) and RStudio version 1.2.5033 (RStudio Inc., Boston, MA, USA).

## 3. Results

### 3.1. Validation Cohorts and Patient Baseline Characteristics

In the validation cohort A from the University of Freiburg (Germany), 61 patients were included with a confident diagnosis of IPF and available serum samples and clinical data (2001–2013) (Figure 1). In the more recent validation in cohort B from Hannover Medical School, 126 patients had a confident diagnosis of IPF and available serum samples (2014–2018) which were used for the CCL18 measurements and genotyping, and of which 101 with complete follow-up data were included in the survival analysis. The median age in both cohorts was similar (68 and 69 years, respectively), with the majority being of male gender (85% and 88%) (Table 1). Both had moderate-to-severe restrictive pattern of pulmonary function testing with a forced vital capacity of predicted (%FVC) of 74% and 75% and a diffusion capacity for carbon monoxide with single-breath method of predicted (%DLCO) of 48% and 47%. Given the recruitment period between 2001–2011 in cohort A, no patient received antifibrotic therapy in this cohort, while in cohort B, 90% of patients included in the survival analysis received antifibrotic therapy (nintedanib *n* = 46 (46.5%), pirfenidone *n* = 43 (43.4%)). Within the cohorts, clinical baseline characteristics were similar based on *rs2015086* genotype, except for baseline %FVC in cohort A, where with the TT-genotype the FVC was higher (79% (62–88)) compared to patients with the CT-genotype (55% (36–68), *p* = 0.014).

### 3.2. Genotyping of the CCL18 rs2015086 SNP and Serum CCL18 Levels

We genotyped the CCL18 *rs2015086* promoter polymorphism in all patients from cohort A (*n* = 61) and all patients from cohort B (*n* = 126). In both cohorts, the homozygous (TT) rs2015086 genotype was most prevalent (52 (85%) and 91 (72.2%)), similar to IPF patients and healthy controls from the derivation cohort in part A of this work [18]. The heterozygous CT-genotype was present in 9 (15%) and 34 (27%) of patients, respectively. Only one patient in cohort B had a homozygous CC genotype.

Median serum CCL18 levels at baseline was 120.7 ng/mL (IQR 82.1–163.4) for cohort A (Figure 2A) and 131 ng/mL (IQR 103–165) for cohort B (Figure 2C). In both cohorts, CCL18 levels were significantly lower in carriers of the TT-*rs2015086* genotype compared to the CT genotype: cohort A, TT-genotype 115.8 ng/mL (IQR 72.4–139.1) vs. CT-genotype 234 ng/mL (IQR 207.3–244.2), *p* = 5 × 10^−6^; cohort B: TT-genotype 120 ng/mL (IQR 89–154) vs. CT-genotype 159.5 (IQR 128–199), *p* = 0.0001 (Figure 2B,D). The one patient with the CC-genotype had a CCL18 level (226 ng/mL) higher than the median values of the TT and CT subgroups. For the subgroup of cohort B used for the survival analysis (*n* = 101; Table 1), the median CCL18 level at baseline was 140 ng/mL (IQR 115–167).

### 3.3. Serum CCL18 Levels under Antifibrotic Therapy

We continued to analyze the effect of antifibrotic treatment in a subset of cohort B (*n* = 119 who received antifibrotics) on CCL18 levels at three different timepoints (Baseline before initiation of antifibrotic therapy; 1st follow-up usually within 4–6 weeks and 2nd follow-up usually between 4–6 months following therapy initiation). At baseline, serum CCL18 was 131.3 ng/mL (IQR 96.7–163) and increased slightly and statistically significantly to 140.8 ng/mL (IQR 114.2–180.1) during follow-up and under antifibrotic therapy (latest available; 1st or 2nd follow-up), *p* = 0.0036 (Figure 3A). The overall trend towards increase of CCL18 levels was independent of the type of drug (pirfenidone or nintedanib; Figure 3B, *p* = 0.207) and from the *rs2015086* genotype (TT vs. CT/CC, *p* = 0.291, Figure 3B,C).

### 3.4. Analysis of Overall and Progression-Free Survival Depending on CCL18 Cutoffs and rs2015085 Genotype

To study the effect of CCL18 levels at baseline on survival, we used the median CCL18 serum-cut-off (CCL18^low^ or CCL18^high^) in each cohort with available survival data in all 61 patients in cohort A (120 ng/mL); and 101/126 patients in cohort B (140 ng/mL). With univariate comparison using a log-rank test, patients with serum CCL18 levels above the cut-off at baseline had worse survival in both groups: in cohort A without antifibrotics, 17 died in the CCL18^high^ group (53%) vs. 9 in the CCL18^low^ (31%), *p* = 0.041 (Figure 4A); in cohort B with antifibrotics, 25 in the CCL18^high^ group died (47.2%) vs. 11 (22.9%) in the CCL18^low^ group, log-rank *p* = 0.003) (Figure 4C).

The *rs2015086* genotype also discriminated survival curves in cohort A, where the CT heterozygous genotype was associated with poor survival; in the CT-group, 7 died (78%) vs. 19 (37%) in the TT genotype group (*p* = 0.0051) (Figure 4B). Interestingly, in cohort B, there was no difference in mortality rates by genotype (36.8% with TT-genotype; 35.5% with CT or CC genotype, log-rank *p* = 0.770) (Figure 4D).

Progression-free survival in cohort B showed similar effects. In the CCL18^high^ group, 47 (88.7%) had progressive disease or died at 36 months vs. 37 (77.1%) in the CCL18^low^ group (*p* = 0.004) (Figure 4E), while there was no statistical difference with the *rs2015086* genotype (80.9% with TT genotype vs. 87.1% with CT or CC genotype, *p* = 0.630, Figure 4F).

In a multivariable Cox regression model for survival (Figure 5A) and progression-free survival (Figure 5B), the CCL18 cut-off of 140 ng/mL was an independent risk factor for both outcomes (adjusted hazard ratio for death at 36 months 2.25 (95% confidence interval 1.08–4.69), *p* = 0.031; adjusted hazard ratio for progression or death at 36 months 1.65 (95 confidence interval 1.08–2.54), *p* = 0.022).

There was no relevant interaction between the *rs2015086* genotype and antifibrotic treatment (nintedanib and pirfenidone), both with regards to survival and progression-free survival (corresponding *p*-values of 0.852 and 0.930 for survival; 0.425 and 0.432 for progression-free survival).

## 4. Discussion

The main finding of this second part of our work is that we have validated the association between the genotype of the CCL18 gene promotor SNP, *rs2015086,* and the CCL18 serum levels in two independent replication cohorts. In addition, we show the predictive value of CCL18 serum levels with respect to outcome in patients treated with antifibrotics. Higher CCL18 baseline serum levels and presence of the heterozygous “CT” *rs2015086* genotype were associated with worse survival in one validation cohort derived from the pre-antifibrotic era (similar to that of the derivation cohort described in Part A of this work), while in the validation cohort with antifibrotics, only increased CCL18 levels implicated a poor prognosis but not the *CCL18* genotype alone. Lastly, this is the first study to our knowledge to report longitudinal changes of CCL18 under antifibrotic therapy, demonstrating that CCL18 levels rise regardless of antifibrotic treatment.

In IPF, CCL18 is mainly produced by alveolar macrophages and a marker of alternative (M2) activation [10,23]. CCL18 reacts as a chemoattractant to T-lymphocytes, epithelial cancer cells and via cross-talk with fibroblasts to upregulate collagen production [10,11]. Its association with poor prognosis has been demonstrated in IPF consistently [11,13], and also in other interstitial lung disease-associated diseases like systemic sclerosis [12,15,24]. Our data thus independently confirms the predictive value of CCL18 at baseline and show that this is irrespective of antifibrotic treatment. Neighbors et al. reported the predictive value of CCL18 using baseline biomarkers from the combined ASCEND and CAPACITY trials for pirfenidone [13], and we now add equal prognostic value for patients treated with nintedanib.

The lack of CCL18 reduction with antifibrotic therapy however is contrary to results from *in-vitro* studies. In murine models and human THP-1 derived macrophages (in-vitro), nintedanib inhibited CCL18 secretion following stimulation with a cocktail of IL-4/IL-13 or IL-4/IL-13 and IL-6 [25]. With pirfenidone, in U-937 derived macrophages CCL18 production stimulated by IL-4 could be partly inhibited by inhibition of STAT6 phosphorylation [26]. Our data does not replicate these findings in human patients. We could also not find antifibrotic therapy (either nintedanib or pirfenidone) to act dependent on baseline CCL18 levels or slope of serial CCL18 levels during therapy, similar to data from Neighbors et al. [13]. Given the diverse in vivo effects of both drugs, it appears that serial CCL18 monitoring does not provide additional value to inform clinical management of patients on antifibrotic therapy.

Unlike cohort A, there was no correlation between clinical outcomes and the *rs2015086* genotype in cohort B. This may be due to the different allele frequencies in the cohorts and potential influence of additional SNPs in the CCL18 gene which were not characterized in these cohorts, given that previously two SNPs of the CCL18 promotor region with functionality have been described (*rs2015086* and *rs712040)* [17]. Perhaps more importantly, there was a considerable overlap between the baseline CCL18 levels between the genotypes in cohort B, while in cohort A CCL18 levels in the CT-genotype group were markedly higher (Figure 2). The *rs2015086* genotype influences CCL18 serum levels, but they are perhaps more reflective of overall disease activity [10], which is well-correlated with the more advanced reduction of pulmonary function tests in cohort B (Table 1). With progressive disease, CCL18 serum levels seem more predictive of clinical outcomes than the *rs2015086* genotype. Obviously, almost universal use of antifibrotic therapy in cohort B and the shorter follow-up period (4 years vs. 3 years) may have also influenced results, so analysis of the genotype in cohorts with longer follow-up may help clarifying the impact in the settings of antifibrotic therapy.

Another drug which recently was brought back to attention in ILD is tocilizumab, an anti-IL6 receptor antibody, which has been investigated in two randomized controlled trials for systemic sclerosis (SSc) [27,28,29,30]. Therein, both FVC decline in SSc-ILD and serum CCL18 levels were reduced significantly with tocilizumab [27]. Given that CCL18 is associated with the progression and mortality in both SSc and IPF [12,14], the association of M2 macrophage phenotype with stat3 signaling in patients suffering from various types of pulmonary fibrosis including IPF may provide a molecular mechanism for CCL18 in fibrotic lung diseases [31]. In a recent study, the effect of IL-6 expression on TFG-ß related signaling pathways (STAT3, Smad3) in human lung fibroblasts derived from IPF patients could be antagonized by tocilizumab [32]. It is thus conceivable that inhibition of TGF-ß related signaling may also downregulate M2-macrophage activation and CCL18 expression. Since nintedanib showed clinical efficacy for both SSc (a disease partly driven by IL-6 signaling) [33,34] and IPF [4], similar effects may be expected from tocilizumab treatment in IPF, but clinical studies are necessary to confirm this hypothesis.

One of the limitations of our study is the missing genotypes in a fraction of patients in cohort B which may affect the conclusions regarding the prognostic implication of the *rs2015086* genotype with antifibrotic therapy. Our studies without Part A would have the significant limitation that only cohorts from one principal investigator would have been analyzed, despite its prospective design. In the context of the data presented in Part A, however, we confirm the findings with two independent prospective cohorts.

In conclusion, serum CCL18 levels correlate with the *CCL18 rs2015086* genotype and predicts mortality and disease progression in two independent prospective cohorts irrespective of antifibrotic treatment.

## Figures and Tables

**Figure 1 jcm-09-01993-f001:**
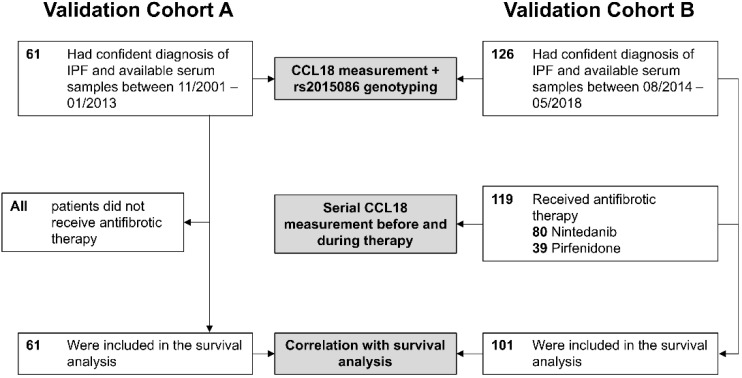
Study population flowchart of the two validation cohorts. CCL18 (Chemokine (C-C motif) ligand 18).

**Figure 2 jcm-09-01993-f002:**
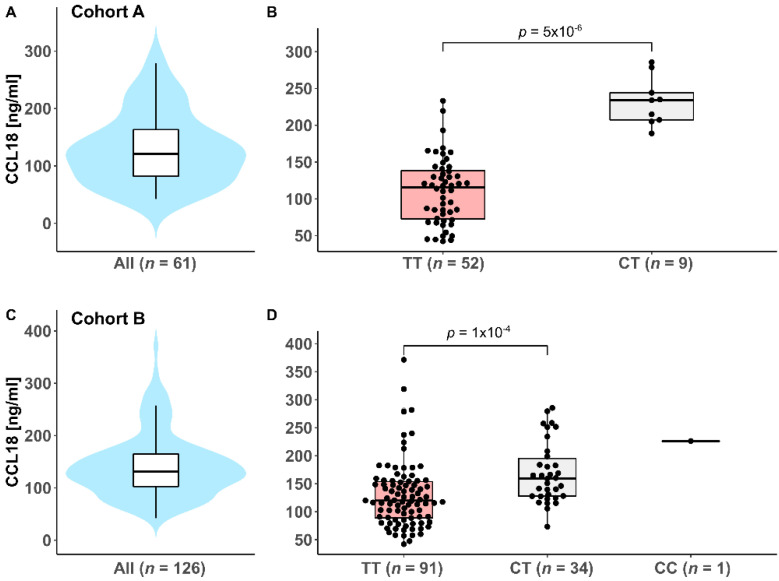
Serum CCL18 measurements at baseline in all patients (**A**,**C**) and dependent on *rs2015086 CCL18*-genotype (**B**,**D**).

**Figure 3 jcm-09-01993-f003:**
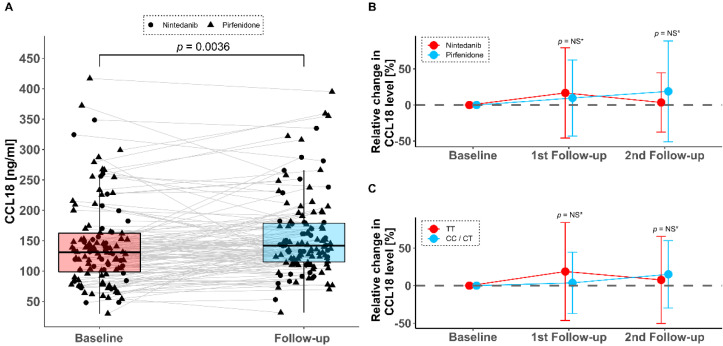
Absolute serum CCL18 levels at baseline in 119 patients in cohort B and during antifibrotic therapy (**A**). The mean relative change from baseline dependent on antifibrotic drug (**B**) and on *rs2015086 CCL18*-genotype (**C**) showed a similar trend in the subgroups. * Wilcoxon test group comparison at individual timepoints; NS = not statistically significant.

**Figure 4 jcm-09-01993-f004:**
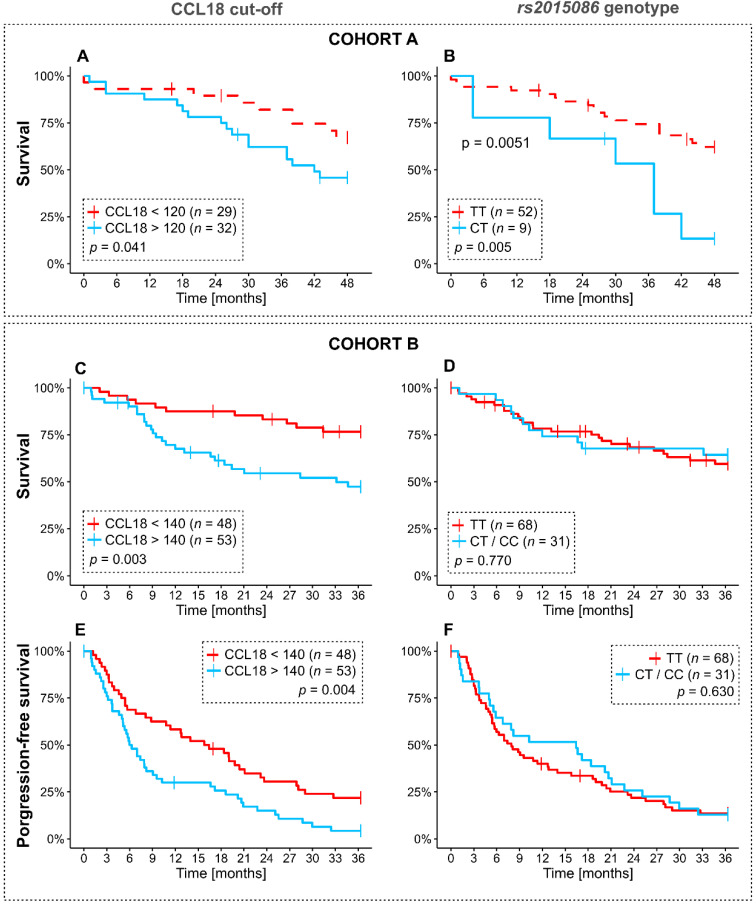
Survival dependent on median CCL18 cutoff in cohort A (120 ng/mL) (**A**) and cohort B (140 ng/mL) (**C**) and survival by *rs2015086* genotype in both cohorts (**B**,**D**). For cohort B, progression-free survival is shown in dependent on 140 ng/mL CCL18 cutoff (**E**) and on *rs2015086* genotype (**F**). *p*-values by log-rank test.

**Figure 5 jcm-09-01993-f005:**
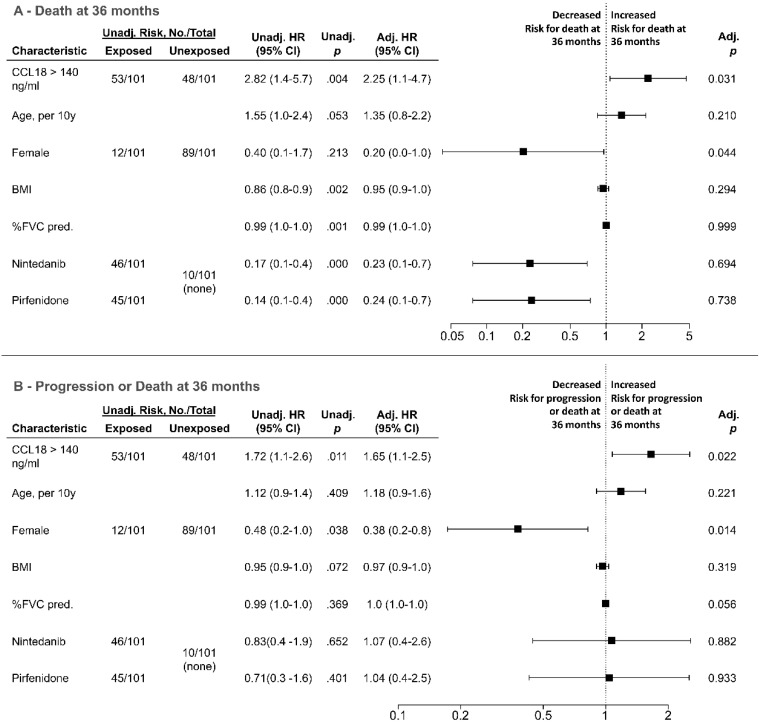
Multivariable cox-regression model for death at 24 months (**A**) and progression or death at 36 months (**B**) in cohort B (*n* = 101 patients). Adj. = adjusted; CI = confidence interval; FVC pred. = forced vital capacity (percent of predicted); HR = hazard ratio; unadj. = unadjusted; y = years.

**Table 1 jcm-09-01993-t001:** CCL18 genotype specific baseline characteristics in patients with idiopathic pulmonary fibrosis used for the survival analysis.

Characteristics	Cohort A (*n* = 61)			Cohort B (*n* = 101)		
rs2015086 Genotype	TT (*n* = 52)	CT (*n* = 9)	*p*	TT (*n* = 68)	CT/CC (*n* = 31)	*p*
Male, *n* (%)	44 (85)	8 (89)	0.739	58 (85)	29 (94)	0.243
Age, years (IQR)	68 (63–76)	67 (58–72)	0.600	69 (61–74)	68 (59–76)	0.963
Former or active smoker, *n* (%)	32 (63)	5 (56)	0.683	47 (69)	21 (68)	0.633
Baseline %FVC predicted, (IQR)	79 (62–88)	55 (36–68)	0.014	65 (53–77)	69 (58–82)	0.341
Baseline %DLCO predicted (IQR)	48 (35–60)	48 (41–58)	0.710	48 (36–58)	48 (36–60)	0.925
Antifibrotic therapy	-					0.669
None	52 (100)	9 (100)		8 (12)	2 (6)	
Nintedanib	-			32 (47)	14 (45)	
Pirfenidone	-			28 (41)	15 (49)	

IQR (interquartile range); %FVC (percent of predicted forced vital capacity); %DLCO (percent of predicted diffusion capacity for carbon monoxide with single breath). Two patients in cohort B did not undergo genotyping but had serial serum CCL18 measurements and were included in the survival analysis.

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
