# Peer review of "Genetic Variation in CCL18 Gene Influences CCL18 Expression and Correlates with Survival in Idiopathic Pulmonary Fibrosis—Part B"

_jcm, 2020, doi:10.3390/jcm9061993_

Round 1
Reviewer 1 Report
This study and another study, Part A, clearly confirmed that CCL18 levels are depend on genotype and have an impact on prognosis of IPF patients.
However, there are discrepancies in the results between Cohort A and B. That is, survival and progression-free survival did not show difference between genotype TT and CT/CC in cohort B though cohort A showed significant difference.
The authors explain that this may be due to the different allele frequencies in the cohorts and potential influence of additional SNPs in the CCL18 gene which were not characterized in these cohorts. I can't really understand this consideration, so please explain it more specifically.
On the other hand, the reason why cohort B could not show the difference on survival and PFS by genotype seems to be elsewhere. I think that Figure 2B and D indicate hints about the cause of the discrepancy on results of survival between cohort A and B. In cohort A, when divided by a CCL18 cutoff value of 120, all samples in genotype CT group had a CCL18 value of more than 120. However, in cohort B, when divided by the CCL18 cutoff value of 140, there are not a few samples with a CCL18 value of 140 or less in the genotype CT/CC group. Moreover, in the genotype TT group in cohort B, many samples had a CCL18 value of 140 or more, and the range of distribution included each CCL18 value of the CT/CC group. In other words, since the difference in the distribution of baseline CCL18 values between genotypes was small in cohort B, it is probable that cohort B could not show a difference in prognosis between genotypes. It seems to be a bigger factor than the difference in genetic factors and observation period.
Please give your thoughts on this point.
Author Response
REVIEWER 1:
>This study and another study, Part A, clearly confirmed that CCL18 levels are depend on genotype and have an impact on prognosis of IPF patients.
>However, there are discrepancies in the results between Cohort A and B. That is, survival and progression-free survival did not show difference between genotype TT and CT/CC in cohort B though cohort A showed significant difference.
>The authors explain that this may be due to the different allele frequencies in the cohorts and potential influence of additional SNPs in the CCL18 gene which were not characterized in these cohorts. I can't really understand this consideration, so please explain it more specifically.
RESPONSE: We thank the reviewer for pointing out this inaccuracy. In previous studies (Modi et al 2006), there were two SNPs of the CCL18 gene described with presumed functionality: rs2015086 and rs712040, of which we only analyzed the first one in PART B of this study. Variance in rs712040 may play a role as well, but was not examined further. We specified this in the discussion part (line 254-256). “This may be due to the different allele frequencies in the cohorts and potential influence of additional SNPs in the CCL18 gene which were not characterized in these cohorts, given that previously two SNPs of the CCL18 promotor region with functionality have been described (rs2015086 and rs712040) [Modi et al. 2006].”
We however agree with the next comment of the reviewer, which we also elaborated in the discussion part (see below).
>On the other hand, the reason why cohort B could not show the difference on survival and PFS by genotype seems to be elsewhere. I think that Figure 2B and D indicate hints about the cause of the discrepancy on results of survival between cohort A and B. In cohort A, when divided by a CCL18 cutoff value of 120, all samples in genotype CT group had a CCL18 value of more than 120. However, in cohort B, when divided by the CCL18 cutoff value of 140, there are not a few samples with a CCL18 value of 140 or less in the genotype CT/CC group. Moreover, in the genotype TT group in cohort B, many samples had a CCL18 value of 140 or more, and the range of distribution included each CCL18 value of the CT/CC group. In other words, since the difference in the distribution of baseline CCL18 values between genotypes was small in cohort B, it is probable that cohort B could not show a difference in prognosis between genotypes. It seems to be a bigger factor than the difference in genetic factors and observation period.
>Please give your thoughts on this point.
RESPONSE: We thank the reviewer for these thoughtful comments and agree that the differences in the distribution of CCL18 values at baseline between cohort A and cohort B may explain why we did not find any association of the genotype with survival in cohort B. As the reviewer correctly pointed out, CCL18 levels are influenced by several factors and the genotype is one of them but may not be the most important one. In previous studies we demonstrated that CCL18 levels reflect fibrotic disease activity and raise over time in many IPF patients (Prasse et al. 2006, AJRCCM and Prasse et al. Arthritis Rheumatism 2007). The overall higher CCL18 values in the TT-genotype group of cohort B may very well reflect the overall more advanced disease state, given that in cohort A, FVC in TT group was 79 vs 55% with CT genotype, while in cohort B, FVC in CT/CC Genotype group was slightly higher than in the TT-genotype group. We added this to the discussion (lines 254-262: “Unlike cohort A, there was no correlation between clinical outcomes and the rs2015086 genotype in cohort B. This may be due to the different allele frequencies in the cohorts and potential influence of additional SNPs in the CCL18 gene which were not characterized in these cohorts, given that previously two SNPs of the CCL18 promotor region with functionality have been described (rs2015086 and rs712040) [Modi et al 2006]. Perhaps more importantly, there was a considerable overlap between the baseline CCL18 levels between the genotypes in cohort B, while in cohort A CCL18 levels in the CT-genotype group were markedly higher (Figure 2). The rs2015086 genotype influences CCL18 serum levels, but they are perhaps more reflective of overall disease activity [Prasse et al 2006], which is well-correlated with the more advanced reduction of pulmonary function tests in cohort B (Table 1). With progressive disease, CCL18 serum levels seem more predictive of clinical outcomes than the rs2015086 genotype.”
Reviewer 2 Report
- Line 145-146 when you say “the homozygous genotype” I believe you are referring to the TT genotype (and not the CC genotype), can you clarify this in the text?
- Line 150 “Median serum CCL18 levels in at baseline…” remove “in”
- Line 155, “The one patient with the CC-genotype had the highest CCL18 level (226 ng/ml).” is misleading as other individual samples have higher CCL18 levels, clarify that you mean the CCL18 level in the one CC sample is higher than the mean levels in the TT or CTs.
- How were CCL18 high and low thresholds set? Why are they different between cohort A and cohort B?
- Very interesting findings! Have you done a treatment by genotype interaction to determine if the efficacy of the treatment is genotype dependent? Figure 4 B versus Figure 4 D seems to indicate that individuals with the CT genotype may benefit from treatment while the TT for not (even with the small numbers)
- Figure 5, what does the p on the far-right side indicate?
Author Response
REVIEWER 2:
>Line 145-146 when you say “the homozygous genotype” I believe you are referring to the TT genotype (and not the CC genotype), can you clarify this in the text?
RESPONSE: We added the reference towards the “TT” Genotype in the text to clarify – thank you for pointing this out.
>Line 150 “Median serum CCL18 levels in at baseline…” remove “in”
RESPONSE: We corrected this typo as indicated.
>Line 155, “The one patient with the CC-genotype had the highest CCL18 level (226 ng/ml).” is misleading as other individual samples have higher CCL18 levels, clarify that you mean the CCL18 level in the one CC sample is higher than the mean levels in the TT or CTs.
RESPONSE: The reviewer is completely right with this suggestions and we changed the wording as suggested: “The one patient with the CC-genotype had a CCL18 level (226 ng/ml) higher than the median values of the TT and CT subgroups. ” (page 5 lines 159-160)
>How were CCL18 high and low thresholds set? Why are they different between cohort A and cohort B?
RESPONSE: The CCL18-cutoff / threshold was set at the median of the respective cohort (120 ng/ml in Cohort A and 140 ng/ml in cohort B). The distribution of CCL18 was quite different between the cohorts, which higher CCL18 levels in cohort B (corresponding to the more advanced disease state in cohort B), and cohort B patients received antifibrotic therapy (unlike cohort A), so we felt setting individual cut-offs for the both cohorts was more appropriate than finding a cutoff in the merged population. We previously described that we used the median CCL18 value in the method section (line 108), but now added that we derived the cut-off per cohort to be more specific (page 3, lines 108-109).
>Very interesting findings! Have you done a treatment by genotype interaction to determine if the efficacy of the treatment is genotype dependent? Figure 4 B versus Figure 4 D seems to indicate that individuals with the CT genotype may benefit from treatment while the TT for not (even with the small numbers)
RESPONSE: This is a very important question, which we have had not elaborated in the manuscript yet and we do agree that the visual comparison of figures 4B vs 4D creates the impression that antifibrotic therapy may be beneficial particularly for patients with the CT genotype. We performed a gene-treatment interaction analysis as previously performed by Oldham / Noth et al. AJRCCM 2015 by entering as variables the rs2015086 genotype (TT vs CT / CC), the antifibrotic treatment modality (none; nintedanib; pirfenidone) and the interaction-term between genotype and antifibrotic drug. The interaction p-values were 0.425 for genotype*nintedanib and 0.432 for genotype*pirfenidone, which is clearly above the recommended cut-off of <0.01 to assume relevant interaction (Smits et al. J Clin Epidemiology 2005). Results were similar for progression-free survival (p-values: nintedanib 0.852; pirfenidone 0.930) Our study was not designed (or powered) to show statistical interactions between the relatively small number of patients treated with antifibrotic agents and genotype, but our data thus does not show an overt interaction between treatment efficacy and rs2015086 genotype. We briefly included this analysis in manuscript, both in the methods and results part: Methods (page 3, lines 112-115): “In analogy to Oldham et al. 2015, rs2015085-genotype-treatment interaction was tested with a multivariable cox regression model including the genotype (TT or CT/CC), antifibrotic agent and genotype-treatment interaction term [Oldham et al 2015; Smits et al. 2005]. Gene-treatment interaction was considered present if the p-value of the Wald z-statistic for the interaction term was <0.01 [Oldham et al 2015].”. Results: (page 6, lines 206-208): “There was no relevant interaction between the rs2015086 genotype and antifibrotic treatment (nintedanib and pirfenidone), both with regards to survival and progression-free survival (corresponding p-values of 0.852 and 0.930 for survival; 0.425 and 0.432 for progression-free survival).”
>Figure 5, what does the p on the far-right side indicate?
RESPONSE: In Figure 5, we report the univariable (unadjusted) hazard ratio (=Unadj. HR) with the associated unadjusted p-value (5th column) as well as the multivariable (adjusted) HR and associated adjusted p-value (which is the p on the far-right side. We see our figure merits from clarification and added the “unadj.” And “adj.” in front of each “p-value” and also added a list of abbreviations in the legend of the figure.
Round 2
Reviewer 1 Report
The discussion has been improved enough.